# The Effects of Transport Stress (Temperature and Vibration) on Blood Biochemical Parameters, Oxidative Stress, and Gill Histomorphology of Pearl Gentian Groupers

Dan Fang [1], Jun Mei [1,2,3,4] , Jing Xie [1,2,3,4,*] and Weiqiang Qiu [1,2,3,4,*]

1   College of Food Science and Technology, Shanghai Ocean University, Shanghai 201306, China;
    m210300819@st.shou.edu.cn (D.F.); jmei@shou.edu.cn (J.M.)
2   National Experimental Teaching Demonstration Center for Food Science and Engineering, Shanghai Ocean
    University, Shanghai 201306, China
3   Shanghai Engineering Research Center of Aquatic Product Processing and Preservation,
    Shanghai 201306, China
4   Shanghai Professional Technology Service Platform on Cold Chain Equipment Performance and Energy
    Saving Evaluation, Shanghai 201306, China
*   Correspondence: jxie@shou.edu.cn (J.X.); wqqiu@shou.edu.cn (W.Q.); Tel.: +86-21-6190-0351 (J.X.);
    +86-21-6190-0368 (W.Q.)

**Abstract:** The transport of living fish is an important part of the fish farming process. The transport usually causes fish stress. This study evaluated the effects of transport temperature and vibration frequency on water quality, blood biochemical parameters, gill histomorphology, oxidative stress, and meat quality of pearl gentian groupers after transport. First, 1-year-old groupers (450 ± 25 g) were transported in plastic bags for 48 h, including the following treatments: no shaking, transported at 15 °C, shaking at 70 rpm, transported at 15 °C (15 °C/70 rpm); shaking at 120 rpm, transported at 15 °C (15 °C/120 rpm); no shaking, transported at 25 °C; shaking at 70 rpm, transported at 25 °C (25 °C/70 rpm); and shaking at 120 rpm, transported at 25 °C (25 °C/120 rpm). Serum, liver, gill, and muscle samples were collected for testing at 0, 12, 24, 36, and 48 h of exposure. During the 48 h transport, total ammonia nitrogen (TAN), superoxide dismutase (SOD), catalase (CAT), glutathione peroxidase (GSH-PX), and malondialdehyde (MDA) were significantly increased in the transport group compared to the control group. In the early stage of transportation, aspartate aminotransferase (AST), alanine aminotransferase (ALT), cortisol (COR), lactate dehydrogenase (LDH), and glucose (GLU) in the transportation groups were significantly higher than those in the control group, while the water quality pH and dissolved oxygen (DO) levels decreased significantly. Compared with untransported fish, the total free amino acid (TFAA) content increased by 40.27% and 31.74% in the 25 °C/70 rpm and 25 °C/120 rpm groups, respectively. In addition, the results of hematoxylin–eosin staining and scanning electron microscopy showed that the epithelial cells in the high-speed group were swollen, the gill lamella was severely curved, and a large amount of mucus was secreted. This study explores the basic information of transportation, which will help to select the conditions that are more suitable for the successful transportation of pearl gentian groupers.

**Keywords:** pearl gentian grouper; stress; transport; temperature; vibration

**Key Contribution:** The optimal transport conditions for pearl gentian groupers were 15 °C/70 rpm; which provides a scientific basis for finding ways to ease the transportation pressure and improve the survival rate of groupers.



## 1. Introduction

The transport of live fish is an integral part of the fish farming industry. However, during transport, fish are stimulated by different stressors, including overcrowding, hypoxia, vibration, water temperature, and ammonia [1,2]. As the transport time increases,

transport stress may reduce fish's physiological functions and resistance to pathogens. Temperature and vibration are environmental factors during the transportation of live fish and are the main stressors that induce transportation stress [3,4]. In addition, fish are in a closed environment, and their metabolic waste can affect water quality parameters such as pH, ammonia, carbon dioxide, and nitrite. The accumulation of total ammonia nitrogen (TAN) in water will cause metabolic disorders and enzyme dysfunction in fish [5].

Water temperature is one of the most important environmental factors in aquaculture. Previous studies have shown that temperature stress can adversely affect fish, including disrupting antioxidant enzyme activities [4], activating apoptosis [6], and reducing disease resistance [7]. Heat stress can also weaken the fish's immune defenses [8], induce oxidative stress [9], and increase reactive oxygen species (ROS) production. Excess ROS can react with membrane lipids to generate malondialdehyde (MDA) [10]. Therefore, MDA is often used as an indicator of oxidative stress.

In the process of closed-water transportation, the vibration caused by land is also one of the environmental factors affecting the stress of organisms. Vibrations cause the fish's body to produce a stress response and cause mechanical damage [11]. The main stress response in fish is mediated by acting on the hypothalamic–pituitary–renal interstitial (HPI) axis, which subsequently induces the internal release of catecholamines and cortisol (COR) [12,13]. Increased COR levels trigger elevated blood glucose levels and tissue damage [14]. Vibrations cause the fish's physiological metabolism to lose balance, leading to intensified metabolic processes in the body, increasing its demand for oxygen and the excretion of metabolites. The transported fish will secrete more mucus, pollute the water quality, and cause the death of fish [15]. Wang et al. [16] found that after largemouth bass were transported at 100 rpm for 12 h, vacuolization and intestinal damage occurred in their hepatocytes.

The pearl gentian grouper (Epinephelus fuscoguttatus ♀× Epinephelus lanceolatus ♂) is a new species of grouper [17]. The grouper is one of the most frequently farmed deep-sea fish species in coastal Asia. It has the advantages of fast growth, strong disease resistance, and delicious meat [18]. In recent years, large-scale intensive farming of pearl gentian groupers has become increasingly common. However, during the transport of fish from fishing grounds to marine farms, changes in the physical and chemical parameters of the transport water can result in strong irritation and pressure on the fish [19]. There is a great need to understand how fish respond to water quality changes and transport stress in order to determine the optimal conditions for transporting live fish [20]. However, few studies have reported the effects of temperature and vibration co-transport stress on gill tissue morphology and oxidative stress in the pearl gentian grouper. Therefore, this study aimed to evaluate the optimal transport temperature and vibration rate of the pearl gentian grouper in terms of water quality parameters, oxidative capacity, blood biochemical parameters, and gill tissue morphology to provide a scientific basis for relieving stress and improving the survival rate of groupers during transportation in the future.

## 2. Materials and Methods

### 2.1. Experimental Fishes

First, 1-year-old pearl gentian groupers (mean weight 450 ± 25 g, mean body length 30.0 ± 2.5 cm) were obtained from the market in Pudong New Area (Shanghai, China). The groupers were placed in a 600 L recirculating water culture pond for a temporary respite of 24 h. All water quality parameters were maintained in the appropriate range: water temperature was 20–23 °C, salinity was 20–25‰, dissolved oxygen was 7–8 mg/L, and fish density was 50 g/L. During cultivation, water was circulated by a recirculating filter, and replaced 50% of the water every 12 h. Groupers fasted before the experiment. This experiment was authorized by the Animal Care and Use Committee of Shanghai Ocean University (SHOU-DW-2022-103).

### 2.2. Simulated Transportation

The experimental fish were subjected to simulated transport after 24 h of temporary rearing. The fish were randomly dispersed in 24 double-layered nylon plastic bags (50 cm × 90 cm, 6 fish in each bag, fish to water ratio was 1: 3, 24 shipping boxes). Each nylon bag was filled with approximately one-third seawater (10 L) and two-thirds oxygen. The nylon bag was sealed with a rubber band. We set the following transport conditions: no shaking, transported at 15 °C; shaking at 70 rpm, transported at 15 °C (15 °C/70 rpm); shaking at 120 rpm, transported at 15 °C (15 °C/120 rpm); no shaking, transported at 25 °C; shaking at 70 rpm, transported at 25 °C (25 °C/70 rpm); and shaking at 120 rpm, transported at 25 °C (25 °C/120 rpm). The fish were then placed on a simulated transport table (Changzhou, China) for 48 h. The control and experimental groups were sampled at 0, 12, 24, 36, and 48 h after the simulated transport. Survival of groupers was measured at each sampling point. Figure 1 illustrates the design process of the experiment.

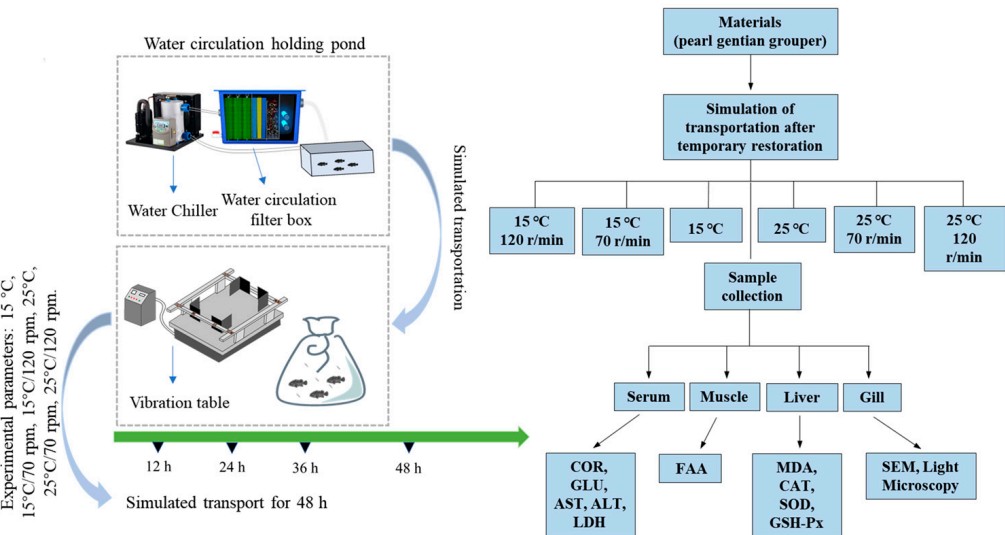

**Figure 1.** Schematic diagram of simulated transport of pearl gentian groupers. The groupers were temporarily raised for 24 h through a chiller and a circulating water tank. The fish were placed in a plastic bag for oxygenation and placed on a vibration table to simulate transportation for 48 h.

### 2.3. Samples Collection

Following the accomplishment of the experiment, five fish were taken from each bag for analysis of blood biochemical parameters, oxidative stress, gill histomorphology, and meat quality. The fish were anesthetized with MS-222 (300 mg/L, Sigma, St. Louis, MO, USA), and blood was collected from the tail vein using a 5 mL non-anticoagulant syringe. The blood was separated by centrifugation (4 °C, 20 min, 4050× $g$) after 12 h. The supernatant (serum) was gathered and preserved at −80 °C until its analysis. After collecting the blood samples, the fish were rapidly sectioned on an ice plate to remove the tissues (gills, liver, and muscle). The liver and the second-gill tissue were placed in 0.85% saline to wash the blood off their surface. The liver was preserved at −80 °C for subsequent manipulation.

### 2.4. Water Quality

Water quality parameters, including pH, dissolved oxygen (DO), and total ammonia nitrogen, were sampled and analyzed immediately after simulated transports of 0, 12, 24, 36, and 48 h. The pH of the transport water was measured with a hand-held portable pH meter (Sartorius, PB-10). DO was determined by a portable dissolved oxygen meter (JPSJ-605F, Shanghai, China). According to Qiang et al. [21], we measured the TAN using the spectrophotometric method.

### 2.5. Serum Biochemical Analysis

Aspartate aminotransferase (AST), alanine aminotransferase (ALT), glucose (GLU), and lactate dehydrogenase (LDH) were measured using commercial kits (Solabra, Beijing, China). Fish Cortisol (COR) ELISA kit was provided by FANKEW (Shanghai, China).

### 2.6. Liver Biochemical Analysis

Superoxide dismutase (SOD), Catalase (CAT), and Malondialdehyde (MDA) were measured using commercial kits (Solabra, Beijing, China). The glutathione peroxidase (GSH-PX) kit was obtained from the Institute of Jiancheng Bioengineering (Nanjing, China).

### 2.7. Light Microscopy Analysis

Gill tissue was collected in 4% paraformaldehyde overnight at 4 °C and rinsed with 0.01 mol/L phosphate-buffered saline (PBS, pH = 7.4). The gills were dehydrated in various gradients of ethanol (70%, 80%, 90%, 95%, and 100%) for 30 min each, then treated with clear xylene and embedded in paraffin. The trimmed paraffin blocks were embedded in a sectioning machine (Leica, RM2135, Nussloch, Germany), and the sections were dewaxed in xylene and various gradients of ethanol and then stained with hematoxylin and eosin (H&E). After rinsing, the sections were sealed with neutral resin [22]. Samples were observed under a light microscope (Olympus BX-43, JESCO, Tokyo, Japan) and photographed. Grouper lamellae were analyzed for length during transport using Image J software.

### 2.8. Scanning Electron Microscopy Analysis

Gill tissue was fixed overnight in 2.5% glutaraldehyde solution before rinsing 3 times with PBS (0.1 mol/L, pH = 7.4) for 15 min each. Tissue samples were then eluted in 30%, 50%, 70%, 80%, 90%, 95%, and 100% ethanol for 15 min. Samples were replaced twice with isoamyl acetate for 15 min. The gill samples were fixed on an electron microscope sample tray and refrigerated overnight at −80 °C. After vacuum drying for 48 h, plating was performed [23]. Scanning electron microscopy (Hitachi SU5000, Tokyo, Japan) was used to observe and photograph the samples.

### 2.9. Free Amino Acids

A total of 2 g of minced meat was homogenized with 10 mL of 5% TCA and centrifuged (10 min, $10,000 \times g$). The extract and centrifugation were repeated, and the supernatant was combined and fixed to 25 mL. The supernatant was filtered through a 0.22 um filter into a brown sampling bottle. Analysis was carried out by means of an automated amino acid analyzer (Hitachi L-8800, Tokyo, Japan).

### 2.10. Statistical Analysis

Values were expressed as mean ± standard deviation (SD). Referring to the method of Slami et al. [24], the independent and interactive effects of vibration frequency, temperature, and transport time were analyzed by three-way ANOVA (Table 1). If the interactions were significant, Duncan's multiple range test was used. Before Duncan's test was applied, the homogeneity of samples was detected by Levene, and the significance level of all samples was $p < 0.05$. SPSS and origin were used for data processing and plotting.

**Table 1.** The interaction results of vibration frequency, temperature, and transport time on FAAs, water quality parameters, blood biochemical parameters, and oxidative stress are shown below.

| Parameters | Source of Variation | df | F | *p*-Value |
|---|---|---|---|---|
| Asp | Vibration | 2 | 113.694 | 0.000 |
| | Temperature | 1 | 891.577 | 0.000 |
| | Time | 2 | 899.496 | 0.000 |
| | Vibration × Temperature | 2 | 229.825 | 0.000 |

**Table 1.** *Cont.*

| Parameters | Source of Variation | df | F | *p*-Value |
|---|---|---|---|---|
| | Vibration × Time | 4 | 158.742 | 0.000 |
| | Temperature × Time | 2 | 69.978 | 0.000 |
| | Vibration × Temperature ×Time | 4 | 351.644 | 0.000 |
| Thr | Vibration | 2 | 1553.037 | 0.000 |
| | Temperature | 1 | 6899.695 | 0.000 |
| | Time | 2 | 9023.207 | 0.000 |
| | Vibration × Temperature | 2 | 1003.614 | 0.000 |
| | Vibration × Time | 4 | 1087.433 | 0.000 |
| | Temperature × Time | 2 | 2951.912 | 0.000 |
| | Vibration × Temperature ×Time | 4 | 2163.354 | 0.000 |
| Ser | Vibration | 2 | 3085.985 | 0.000 |
| | Temperature | 1 | 5251.312 | 0.000 |
| | Time | 2 | 9838.697 | 0.000 |
| | Vibration × Temperature | 2 | 270.116 | 0.000 |
| | Vibration × Time | 4 | 1279.598 | 0.000 |
| | Temperature × Time | 2 | 3896.565 | 0.000 |
| | Vibration × Temperature ×Time | 4 | 409.435 | 0.000 |
| Glu | Vibration | 2 | 1102.377 | 0.000 |
| | Temperature | 1 | 8871.22 | 0.000 |
| | Time | 2 | 7567.827 | 0.000 |
| | Vibration × Temperature | 2 | 667.828 | 0.000 |
| | Vibration × Time | 4 | 451.975 | 0.000 |
| | Temperature × Time | 2 | 299.909 | 0.000 |
| | Vibration × Temperature ×Time | 4 | 499.321 | 0.000 |
| Gly | Vibration | 2 | 17,603.247 | 0.000 |
| | Temperature | 1 | 478,216.287 | 0.000 |
| | Time | 2 | 38,773.851 | 0.000 |
| | Vibration × Temperature | 2 | 3022.137 | 0.000 |
| | Vibration × Time | 4 | 9400.451 | 0.000 |
| | Temperature × Time | 2 | 14,287.42 | 0.000 |
| | Vibration × Temperature ×Time | 4 | 8589.165 | 0.000 |
| Cys | Vibration | 2 | 174.501 | 0.000 |
| | Temperature | 1 | 37,580.373 | 0.000 |
| | Time | 2 | 10,097.668 | 0.000 |
| | Vibration × Temperature | 2 | 150.553 | 0.000 |
| | Vibration × Time | 4 | 201.304 | 0.000 |
| | Temperature × Time | 2 | 10,176.188 | 0.000 |
| | Vibration × Temperature ×Time | 4 | 166.22 | 0.000 |
| Val | Vibration | 2 | 303.916 | 0.000 |
| | Temperature | 1 | 54,920.23 | 0.000 |
| | Time | 2 | 15,503.886 | 0.000 |
| | Vibration × Temperature | 2 | 311.842 | 0.000 |
| | Vibration × Time | 4 | 354.428 | 0.000 |
| | Temperature × Time | 2 | 15,252.433 | 0.000 |
| | Vibration × Temperature ×Time | 4 | 331.264 | 0.000 |
| Ile | Vibration | 2 | 336.723 | 0.000 |
| | Temperature | 1 | 61,708.064 | 0.000 |
| | Time | 2 | 19,835.759 | 0.000 |
| | Vibration × Temperature | 2 | 365.589 | 0.000 |
| | Vibration × Time | 4 | 469.432 | 0.000 |
| | Temperature × Time | 2 | 17,646.533 | 0.000 |
| | Vibration × Temperature ×Time | 4 | 441.031 | 0.000 |

**Table 1.** *Cont.*

| Parameters | Source of Variation | df | F | *p*-Value |
|---|---|---|---|---|
| Leu | Vibration | 2 | 51.933 | 0.000 |
| | Temperature | 1 | 145,997.2 | 0.000 |
| | Time | 2 | 3393.513 | 0.000 |
| | Vibration × Temperature | 2 | 180.624 | 0.000 |
| | Vibration × Time | 4 | 2192.068 | 0.000 |
| | Temperature × Time | 2 | 41,674.346 | 0.000 |
| | Vibration × Temperature ×Time | 4 | 1752.751 | 0.000 |
| Tyr | Vibration | 2 | 6.475 | 0.004 |
| | Temperature | 1 | 2015.841 | 0.000 |
| | Time | 2 | 11.41 | 0.000 |
| | Vibration × Temperature | 2 | 3.841 | 0.031 |
| | Vibration × Time | 4 | 43.12 | 0.000 |
| | Temperature × Time | 2 | 601.765 | 0.000 |
| | Vibration × Temperature ×Time | 4 | 3.813 | 0.011 |
| Phe | Vibration | 2 | 318.167 | 0.000 |
| | Temperature | 1 | 36,172.254 | 0.000 |
| | Time | 2 | 233.726 | 0.000 |
| | Vibration × Temperature | 2 | 7.608 | 0.002 |
| | Vibration × Time | 4 | 850.75 | 0.000 |
| | Temperature × Time | 2 | 3519.479 | 0.000 |
| | Vibration × Temperature ×Time | 4 | 64.426 | 0.000 |
| Lys | Vibration | 2 | 150 | 0.000 |
| | Temperature | 1 | 12,904.795 | 0.000 |
| | Time | 2 | 8375.469 | 0.000 |
| | Vibration × Temperature | 2 | 96.678 | 0.000 |
| | Vibration × Time | 4 | 63.197 | 0.000 |
| | Temperature × Time | 2 | 7616.867 | 0.000 |
| | Vibration × Temperature ×Time | 4 | 96.288 | 0.000 |
| His | Vibration | 2 | 1616.965 | 0.000 |
| | Temperature | 1 | 134,482.415 | 0.000 |
| | Time | 2 | 25,560.675 | 0.000 |
| | Vibration × Temperature | 2 | 379.78 | 0.000 |
| | Vibration × Time | 4 | 681.972 | 0.000 |
| | Temperature × Time | 2 | 14,767.69 | 0.000 |
| | Vibration × Temperature ×Time | 4 | 1506.107 | 0.000 |
| Arg | Vibration | 2 | 144.023 | 0.000 |
| | Temperature | 1 | 21,477.223 | 0.000 |
| | Time | 2 | 6785.463 | 0.000 |
| | Vibration × Temperature | 2 | 136.79 | 0.000 |
| | Vibration × Time | 4 | 141.608 | 0.000 |
| | Temperature × Time | 2 | 6627.54 | 0.000 |
| | Vibration × Temperature ×Time | 4 | 147.979 | 0.000 |
| Ph | Vibration | 2 | 31.746 | 0.000 |
| | Temperature | 1 | 2.75 | 0.102 |
| | Time | 4 | 96.711 | 0.000 |
| | Vibration × Temperature | 2 | 3.223 | 0.047 |
| | Vibration × Time | 8 | 7.936 | 0.000 |
| | Temperature × Time | 4 | 18.078 | 0.000 |
| | Vibration × Temperature ×Time | 8 | 0.792 | 0.612 |
| DO | Vibration | 2 | 84.247 | 0.000 |
| | Temperature | 1 | 1417.282 | 0.000 |
| | Time | 4 | 237.357 | 0.000 |
| | Vibration × Temperature | 2 | 14.784 | 0.000 |
| | Vibration × Time | 8 | 5.941 | 0.000 |

**Table 1.** *Cont.*

| Parameters | Source of Variation | df | F | *p*-Value |
|---|---|---|---|---|
| | Temperature × Time | 4 | 24.497 | 0.000 |
| | Vibration × Temperature ×Time | 8 | 3.806 | 0.001 |
| TAN | Vibration | 2 | 420.276 | 0.000 |
| | Temperature | 1 | 272.822 | 0.000 |
| | Time | 4 | 1141.493 | 0.000 |
| | Vibration × Temperature | 2 | 0.329 | 0.721 |
| | Vibration × Time | 8 | 35.096 | 0.000 |
| | Temperature × Time | 4 | 14.454 | 0.000 |
| | Vibration × Temperature ×Time | 8 | 1.141 | 0.350 |
| AST | Vibration | 2 | 62.166 | 0.000 |
| | Temperature | 1 | 1092.343 | 0.000 |
| | Time | 4 | 179.605 | 0.000 |
| | Vibration × Temperature | 2 | 8.46 | 0.001 |
| | Vibration × Time | 8 | 22.458 | 0.000 |
| | Temperature × Time | 4 | 31.339 | 0.000 |
| | Vibration × Temperature ×Time | 8 | 10.941 | 0.000 |
| ALT | Vibration | 2 | 51.675 | 0.000 |
| | Temperature | 1 | 45.713 | 0.000 |
| | Time | 4 | 24.17 | 0.000 |
| | Vibration × Temperature | 2 | 2.749 | 0.072 |
| | Vibration × Time | 8 | 7.035 | 0.000 |
| | Temperature × Time | 4 | 8.578 | 0.000 |
| | Vibration × Temperature ×Time | 8 | 3.33 | 0.003 |
| COR | Vibration | 2 | 240.539 | 0.000 |
| | Temperature | 1 | 1250.821 | 0.000 |
| | Time | 4 | 521.194 | 0.000 |
| | Vibration × Temperature | 2 | 15.784 | 0.000 |
| | Vibration × Time | 8 | 30.728 | 0.000 |
| | Temperature × Time | 4 | 36.169 | 0.000 |
| | Vibration × Temperature ×Time | 8 | 14.087 | 0.000 |
| GLU | Vibration | 2 | 1040.98 | 0.000 |
| | Temperature | 1 | 224.507 | 0.000 |
| | Time | 4 | 1209.673 | 0.000 |
| | Vibration × Temperature | 2 | 35.114 | 0.000 |
| | Vibration × Time | 8 | 279.811 | 0.000 |
| | Temperature × Time | 4 | 69.353 | 0.000 |
| | Vibration × Temperature ×Time | 8 | 66.185 | 0.000 |
| LDH | Vibration | 2 | 248.767 | 0.000 |
| | Temperature | 1 | 1420.729 | 0.000 |
| | Time | 4 | 383.998 | 0.000 |
| | Vibration × Temperature | 2 | 1.503 | 0.231 |
| | Vibration × Time | 8 | 43.575 | 0.000 |
| | Temperature × Time | 4 | 67.161 | 0.000 |
| | Vibration × Temperature ×Time | 8 | 11.214 | 0.000 |
| SOD | Vibration | 2 | 297.552 | 0.000 |
| | Temperature | 1 | 210.732 | 0.000 |
| | Time | 4 | 122.229 | 0.000 |
| | Vibration × Temperature | 2 | 17.527 | 0.000 |
| | Vibration × Time | 8 | 34.359 | 0.000 |
| | Temperature × Time | 4 | 7.015 | 0.000 |
| | Vibration × Temperature ×Time | 8 | 21.037 | 0.000 |
| GSH-PX | Vibration | 2 | 309.374 | 0.000 |
| | Temperature | 1 | 26.216 | 0.000 |
| | Time | 4 | 184.179 | 0.000 |
| | Vibration × Temperature | 2 | 6.244 | 0.003 |

**Table 1.** *Cont.*

| Parameters | Source of Variation | df | F | *p*-Value |
|---|---|---|---|---|
| | Vibration × Time | 8 | 22.767 | 0.000 |
| | Temperature × Time | 4 | 2.515 | 0.051 |
| | Vibration × Temperature ×Time | 8 | 0.634 | 0.746 |
| MDA | Vibration | 2 | 1464.854 | 0.000 |
| | Temperature | 1 | 50.881 | 0.000 |
| | Time | 4 | 981.196 | 0.000 |
| | Vibration × Temperature | 2 | 8.474 | 0.001 |
| | Vibration × Time | 8 | 132.411 | 0.000 |
| | Temperature × Time | 4 | 19.312 | 0.000 |
| | Vibration × Temperature ×Time | 8 | 18.65 | 0.000 |

## 3. Results

### 3.1. Survival Rate

The effect of different temperatures and vibration rates on grouper survival rates is shown in Table 2. The survival rate of the grouper decreased with the increase in speed and temperature. There was only a 33% survival of groupers in the 25 °C/120 rpm group, and the survival rate of groupers was significantly higher in the low-temperature group than in the high-temperature group. In the later stages of transport, groupers began to turn white in color, became agitated, and increased mucus secretion on the body surface.

**Table 2.** Survival of pearl gentian groupers at different temperatures and vibration rates.

| Samples | Keeping Alive Time/h | | | | |
|---|---|---|---|---|---|
| | 0 h | 12 h | 24 h | 36 h | 48 h |
| 15 °C | 100 | 100 | 100 | 100 | 100 |
| 15 °C/70 rpm | 100 | 100 | 100 | 100 | 83 |
| 15 °C/120 rpm | 100 | 100 | 100 | 100 | 67 |
| 25 °C | 100 | 100 | 100 | 100 | 83 |
| 25 °C/70 rpm | 100 | 100 | 100 | 100 | 67 |
| 25 °C/120 rpm | 100 | 100 | 100 | 100 | 33 |

### 3.2. Water Quality Parameters

The water quality parameters are shown in Figure 2. The overall pH value showed a trend of decreasing first and then increasing. During the 24 h transportation, the pH values of the two experimental groups at different temperatures were significantly lower than those of the control group ($p < 0.05$). The DO level in the low-temperature group (15 °C) was still saturated. However, the DO level in the high-temperature group (25 °C) was significantly lower after transport than before transport ($p < 0.05$). In addition, the TAN content increased with the increase in temperature and rotational speed. The concentration of TAN after transportation was significantly higher than that before transportation ($p < 0.05$).

### 3.3. Serum Biochemical Parameters

The effects of different transport temperatures and vibration rates on blood biochemical parameters are shown in Figure 3. The change trends of AST, ALT, and COR in the transportation group were roughly the same; they all increased first and then decreased. During the 36 h of transportation, the AST of the transportation treatment group was significantly increased compared with the control group ($p < 0.05$). Except for 0 h, the ALT activity of the transport group was higher than that of the control group at every sampling point, and the highest values appeared at 24 and 12 h, respectively.

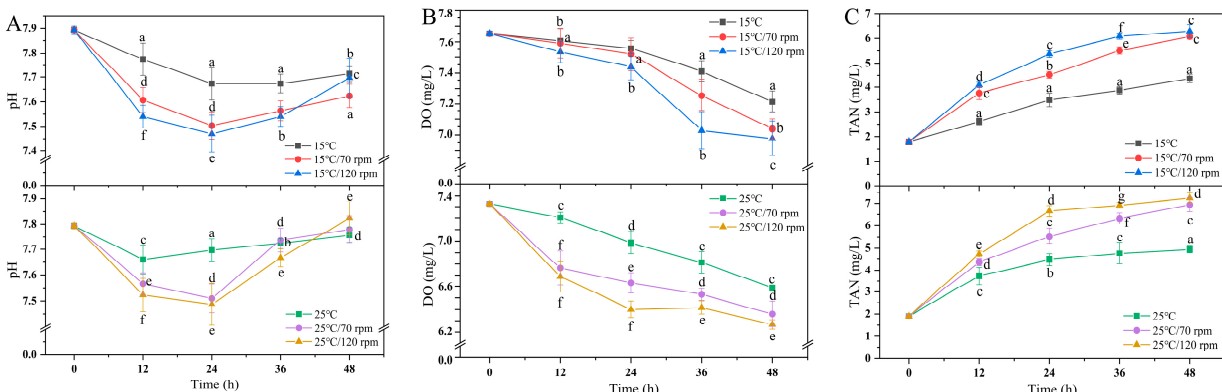

**Figure 2.** Changes in (**A**) water pH, (**B**) DO, and (**C**) TAN during transport. Values were obtained from average of all samples and given as means ± SD (n = 5). Different letters indicate significant differences among the treatments.

**Figure 3.** Changes in (**A**) serum AST, (**B**) ALT, (**C**) COR, (**D**) GLU, and (**E**) LDH during transport. Values were obtained from average of all samples and given as means ± SD (n = 5). Different letters above the bars indicate significant differences among the treatments.

The COR content of the low-temperature group was positively correlated with the vibration rate. During the 48 h of transportation, the serum COR content of the transported fish in the low-temperature group was significantly higher than that in the control group ($p < 0.05$). The COR content in the 25 °C/120 rpm transport group reached its peak at 12 h and gradually decreased after 24 h. During the 12 h of transportation, the GLU content of the high-temperature group was significantly higher than that of the low-temperature group ($p < 0.05$). In addition, during the entire transportation process, the LDH level showed an overall trend of first increasing and then decreasing, and then gradually returning to the pre-stress level. At 12 and 36 h of transportation, the LDH content of the transportation group was significantly higher than that of the control group ($p < 0.05$).

The effects of different transport temperatures and vibration speeds on liver oxidative stress parameters are shown in Figure 4. The overall trend of liver SOD and CAT activities was first increased and then decreased. The activities of SOD, GSH-PX, and CAT in the transport group were significantly higher than those in the control group at 24, 36, and 48 h of transport ($p < 0.05$). At the initial stage of transportation, the SOD activity of the high-speed group was higher than that of the low-speed group, and the GSH-PX activity of the high-temperature group was higher than that of the low-temperature group. In addition, during the 24 h transportation process, the MDA content increased with the transportation time and vibration rate. Similarly, during the 48 h transportation process, the MDA level of the transportation group was significantly higher than that of the control group ($p < 0.05$).

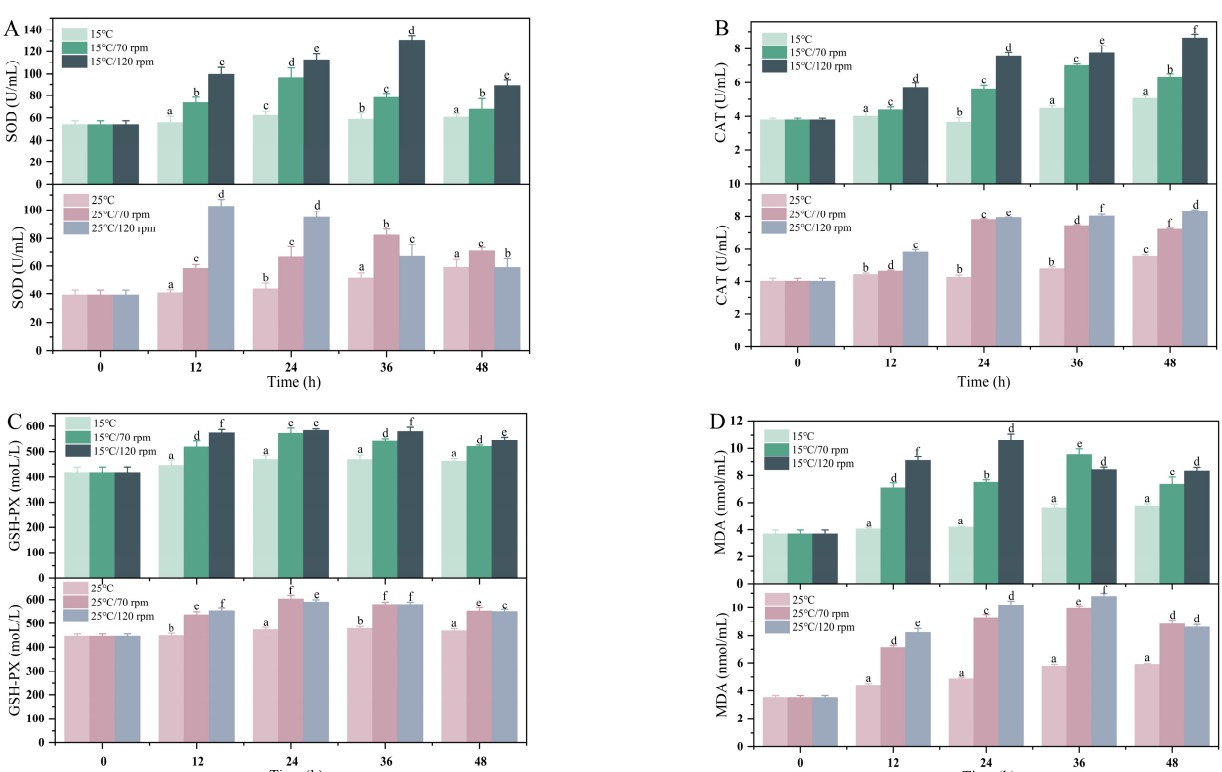

**Figure 4.** Changes in (**A**) liver SOD, (**B**) CAT, (**C**) GSH-PX, and (**D**) MDA during transport. Values were obtained from average of all samples and given as means ± SD (n = 5). Different letters above the bars indicate significant differences among the treatments.

### 3.4. Light Microscopy Analysis

The results of the different transport temperatures and vibration speeds on the gill tissue sections are presented in Figure 5. In the 15 °C and 25 °C controls, the gill lamellae were slender, tightly arranged, evenly distributed, and had normal tissue morphology. The epithelial cells were undamaged. The mitochondria-rich cells were moderately abundant

and were mainly concentrated at the bottom of the gill lamellae. The gill lamellae showed different degrees of damage as the transport time increased. The gill lamellae of the control group were slightly bent, and the structure had no obvious change after being transported for 24 h. Compared to the control group, the gill lamellae in both the lower- and higher-temperature transport groups began to fold. The epithelial cells became swollen, and the distance between the gill lamellae shortened. In the higher-frequency groups (15 °C/120 rpm and 25 °C/120 rpm), the gill lamellae were severely bent, and the apical area was bulbous and thickened. The erythrocytes were unevenly distributed and spread towards the top of the gill lamellae, with no significant changes at the bottom. After 48 h of transport, almost all gill lamellae showed varying degrees of curvature, thickening, and swelling of the epithelial cells. During transport, the majority of the gill lamellae were thickened and curved, rod or S-shaped, with increased mucus cells, and a small number of epithelial cells in the gill lamellae were disrupted and lost. In addition, the length of the lamellae of the pearl gentian grouper was measured (Figure 6). As water temperature and speed increased, the gill lamella length of grouper gills in the treatment group became progressively shorter.

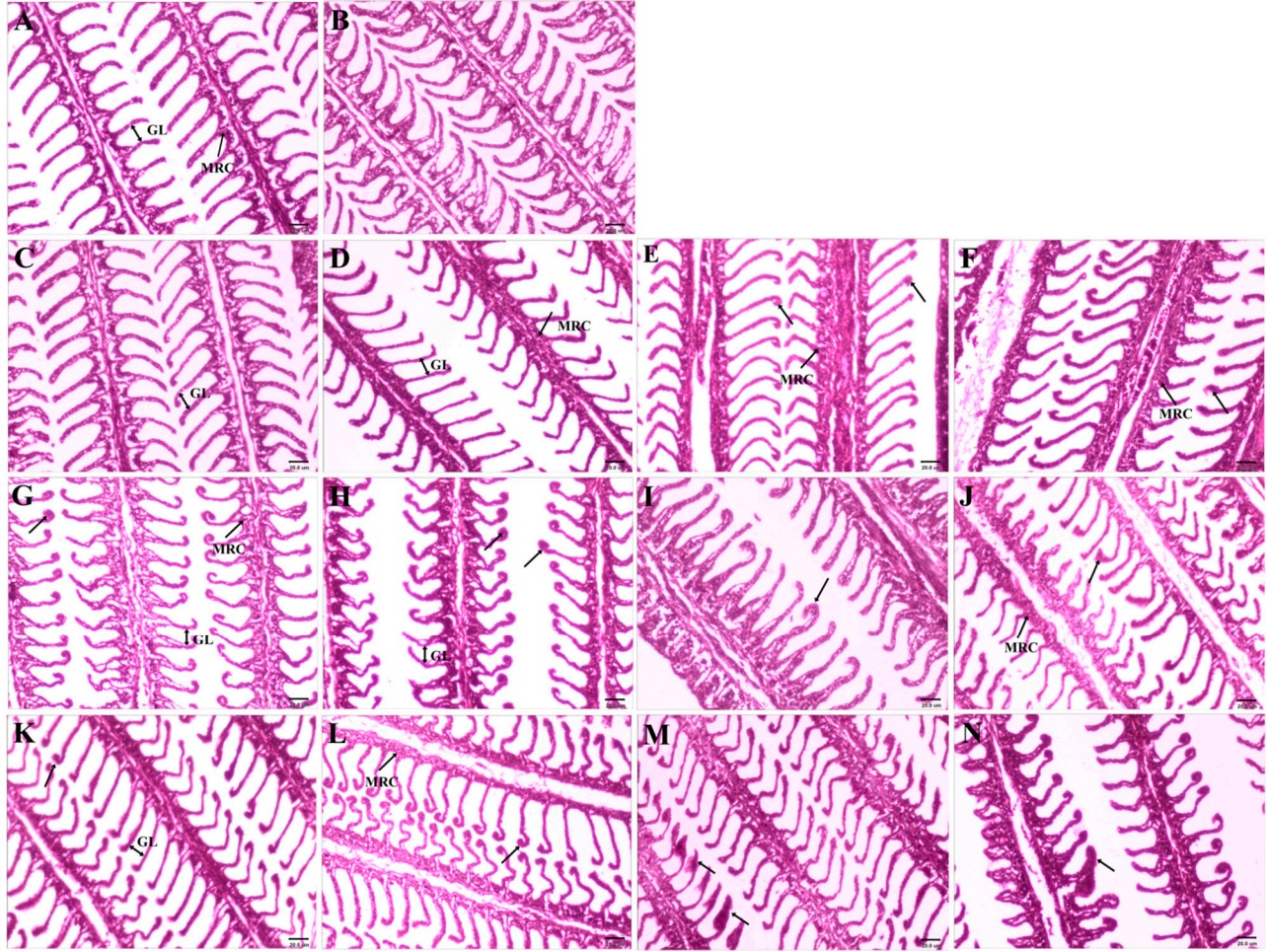

**Figure 5.** Representative H&E-stained micrographs of the gill structure of the pearl gentian grouper at different transport temperatures and vibration rates. (**A**) 15 °C, (**B**) 25 °C, (**C**) 24 h—5 °C, (**D**) 24 h—25 °C, (**E**) 24 h—15 °C/70 rpm, (**F**) 24 h—25 °C/70 rpm, (**G**) 24 h—15 °C/120 rpm, (**H**) 24 h—25 °C/120 rpm, (**I**) 48 h—15 °C, (**J**) 48 h—25 °C, (**K**) 48 h—15 °C/70 rpm, (**L**) 48 h—25 °C/70 rpm, (**M**) 48 h—15 °C/120 rpm, and (**N**) 48 h—25 °C/120 rpm. Bar = 20 μm. Gill lamella (GL), mitochondrial-rich cell (MRC), and epithelial cell swelling (↑).

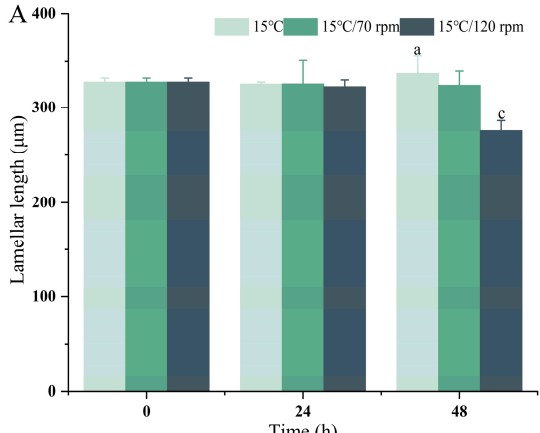
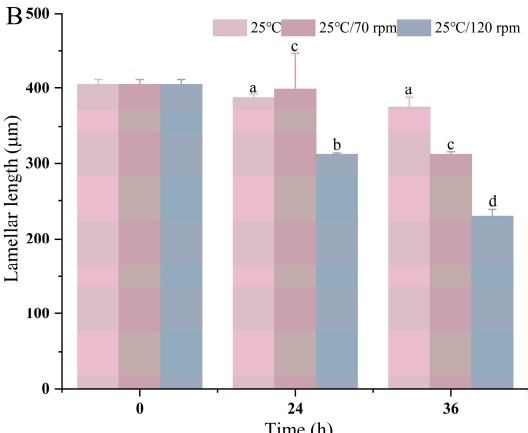

**Figure 6.** Quantification of fish gill structure. Length of gill lamellae of groupers at 15 (**A**) and 25 °C (**B**). Values were obtained from average of all samples and given as means ± SD (n = 5). Different letters above the bars indicate significant differences among the treatments.

### 3.5. Scanning Electron Microscope Analysis

Scanning electron microscope was used to observe the changes in gill tissue (Figure 7). In the control group, the gill structure was complete, the gill sheets were arranged neatly, and the gill surface structure was smooth. After 24 h of transport, the gill tissue of the control group showed little change, while the gill lamellae of the transport treatment group showed folded and curved changes. Gill lamellae in the 25 °C/70 rpm and 15 °C/120 rpm groups began to produce secretory granules and mucus. After 48 h of transport, the gill lamellae of the control group also showed different degrees of damage. The gill lamellae of the 15 °C/70 rpm group showed a large amount of secretion and mucus, and the gill lamellae of the 25 °C/70 rpm group were disorderly. As the speed and temperature increased, the gill lamellae began to adhere to each other and break off.

### 3.6. FAAs Analysis

The effects of different transport temperatures and vibration speeds on the FAA parameters are shown in Table 3. A total of 14 FAAs were detected in the grouper. The main FAAs in fresh groupers were glycine (Gly) and lysine (Lys). The total free amino acid (TFAA) content of most samples was significantly different during transport. Most of the transport groups showed a decreasing trend in TFAA after 48 h of transport. In contrast, the TFAA content increased by 40.27% and 31.74% in the 25 °C/70 rpm and 25 °C/120 rpm groups, respectively, compared to the control group. Figure 8 shows the effects of 48 h of transportation on sweet amino acids (SAA), bitter amino acids (BAA), and umami amino acids (UAA) in grouper muscle. UAA increased by 36.75% and 49.19% in the 25 °C/70 rpm and 25 °C/120 rpm groups, respectively. SAA increased by 68.60% and 51.04%, respectively.

**Table 3.** Effects of temperatures and vibration rates on the free amino acids of the pearl gentian grouper.

| Indicators | Samples | Simulated Transport Time (h) | | |
|---|---|---|---|---|
| | | 0 | 24 | 48 |
| Aspartic Acid [1] (Asp) (mg/kg) | 15 °C | | 0.83 ± 0.03 [b] | 0.98 ± 0.04 [b] |
| | 15 °C/70 rpm | 1.80 ± 0.02 | 0.63 ± 0.01 [a] | 1.48 ± 0.04 [c] |
| | 15 °C/120 rpm | | 0.66 ± 0.02 [a] | 0.83 ± 0.01 [b] |
| | 25 °C | | 1.11 ± 0.02 [c] | 0.94 ± 0.02 [b] |
| | 25 °C/70 rpm | 2.07 ± 0.09 | 1.99 ± 0.04 [f] | 0.85 ± 0.06 [b] |
| | 25 °C/120 rpm | | 1.29 ± 0.03 [d] | 2.94 ± 0.18 [f] |

**Table 3.** *Cont.*

| Indicators | Samples | Simulated Transport Time (h) | | |
|---|---|---|---|---|
| | | 0 | 24 | 48 |
| Threonine [2] (Thr) (mg/kg) | 15 °C | | 4.8 ± 0.05 [a] | 6.57 ± 0.06 [b] |
| | 15 °C/70 rpm | 11.53 ± 0.23 | 6.62 ± 0.07 [b] | 13.51 ± 0.06 [e] |
| | 15 °C/120 rpm | | 3.66 ± 0.01 | 6.11 ± 0.12 |
| | 25 °C | | 9.33 ± 0.13 [c] | 5.11 ± 0.11 [a] |
| | 25 °C/70 rpm | 15.46 ± 0.03 | 15.43 ± 0.24 [f] | 5.35 ± 0.23 [e] |
| | 25 °C/120 rpm | | 8.55 ± 0.03 | 13.46 ± 0.05 |
| Serine [2] (Ser) (mg/kg) | 15 °C | | 9.41 ± 0.14 [a] | 14.54 ± 0.17 [a] |
| | 15 °C/70 rpm | 23.64 ± 0.09 | 16.44 ± 0.08 [c] | 25.47 ± 0.36 [f] |
| | 15 °C/120 rpm | | 6.37 ± 0.09 | 14.15 ± 0.10 |
| | 25 °C | | 24.69 ± 0.17 [d] | 12.62 ± 0.05 [a] |
| | 25 °C/70 rpm | 28.36 ± 0.24 | 26.97 ± 0.55 [e] | 17.92 ± 0.36 [c] |
| | 25 °C/120 rpm | | 14.44 ± 0.09 [b] | 16.66 ± 0.28 [b] |
| Glutamic Acid [1] (Glu) (mg/kg) | 15 °C | | 15.68 ± 0.19 [c] | 20.33 ± 0.24 [b] |
| | 15 °C/70 rpm | 17.66 ± 0.39 | 12.81 ± 0.04 [b] | 30.09 ± 0.38 [d] |
| | 15 °C/120 rpm | | 12.11 ± 0.08 [a] | 19.37 ± 0.27 [a] |
| | 25 °C | | 14.35 ± 0.04 [b] | 26.35 ± 0.21 [c] |
| | 25 °C/70 rpm | 23.91 ± 0.17 | 24.78 ± 0.66 [f] | 36.47 ± 0.07 [e] |
| | 25 °C/120 rpm | | 19.5 ± 0.08 [d] | 37.78 ± 0.43 [e] |
| Glycine [2] (Gly) (mg/kg) | 15 °C | | 29.53 ± 0.14 | 102.87 ± 0.12 [b] |
| | 15 °C/70 rpm | 75.26 ± 0.84 | 47.74 ± 0.09 | 114.55 ± 0.25 |
| | 15 °C/120 rpm | | 22.82 ± 0.04 | 82.58 ± 0.41 [a] |
| | 25 °C | | 159.72 ± 0.15 [d] | 114.57 ± 0.24 [b] |
| | 25 °C/70 rpm | 168.91 ± 0.61 | 180 ± 0.38 | 193.5 ± 0.42 [e] |
| | 25 °C/120 rpm | | 93.35 ± 0.02 [b] | 169.69 ± 0.19 [d] |
| Cysteine (Cys) (mg/kg) | 15 °C | | 9.46 ± 0.04 [b] | 1.04 ± 0.01 [e] |
| | 15 °C/70 rpm | 16.45 ± 0.41 | 15.03 ± 0.06 [c] | 0.74 ± 0.01 [b] |
| | 15 °C/120 rpm | | 9.75 ± 0.05 | 0.87 ± 0.00 [c] |
| | 25 °C | | 0.69 ± 0.01 | 0.96 ± 0.00 [d] |
| | 25 °C/70 rpm | 0.79 ± 0.02 | 0.83 ± 0.00 | 0.77 ± 0.00 [b] |
| | 25 °C/120 rpm | | 0.48 ± 0.01 | 0.66 ± 0.02 [a] |
| Valine (Val)(mg/kg) | 15 °C | | 84.45 ± 0.54 [a] | 7.83 ± 0.00 |
| | 15 °C/70 rpm | 151.96 ± 3.04 | 144.28 ± 0.59 [d] | 5.43 ± 0.00 |
| | 15 °C/120 rpm | | 91.51 ± 0.86 | 6.84 ± 0.02 |
| | 25 °C | | 7.46 ± 0.01 [f] | 7.34 ± 0.01 [e] |
| | 25 °C/70 rpm | 7.35 ± 0.01 | 8.16 ± 0.01 [a] | 6.34 ± 0.02 [c] |
| | 25 °C/120 rpm | | 7.92 ± 0.11 [d] | 7.03 ± 0.02 |
| Isoleucine [3] (Ile) (mg/kg) | 15 °C | | 24.45 ± 0.01 [c] | 5.29 ± 0.02 |
| | 15 °C/70 rpm | 41.61 ± 0.70 | 39.26 ± 0.14 [d] | 3.94 ± 0.06 |
| | 15 °C/120 rpm | | 24.96 ± 0.21 | 4.91 ± 0.01 |
| | 25 °C | | 6.69 ± 0.01 [a] | 4.34 ± 0.06 [c] |
| | 25 °C/70 rpm | 5.60 ± 0.03 | 6.41 ± 0.03 [a] | 4.5 ± 0.03 [d] |
| | 25 °C/120 rpm | | 5.95 ± 0.01 | 5.4 ± 0.04 [e] |
| Leucine [3] (Leu) (mg/kg) | 15 °C | | 2.72 ± 0.01 [a] | 8.66 ± 0.00 [f] |
| | 15 °C/70 rpm | 4.53 ± 0.08 | 4.24 ± 0.00 [b] | 6.2 ± 0.01 [a] |
| | 15 °C/120 rpm | | 2.74 ± 0.04 | 7.93 ± 0.04 [d] |
| | 25 °C | | 10.83 ± 0.01 [a] | 6.89 ± 0.02 [b] |
| | 25 °C/70 rpm | 8.89 ± 0.01 | 10.59 ± 0.01 | 7.35 ± 0.02 [c] |
| | 25 °C/120 rpm | | 9.58 ± 0.01 [b] | 8.71 ± 0.01 [f] |

**Table 3.** *Cont.*

| Indicators | Samples | Simulated Transport Time (h) | | |
|---|---|---|---|---|
| | | **0** | **24** | **48** |
| Tyrosine [3] (Tyr) (mg/kg) | 15 °C | | 1.81 ± 0.00 | 3.77 ± 0.00 [e] |
| | 15 °C/70 rpm | 2.12 ± 0.05 | 1.99 ± 0.01 | 2.97 ± 0.02 [a] |
| | 15 °C/120 rpm | | 1.77 ± 0.02 | 3.57 ± 0.31 [d] |
| | 25 °C | | 4.6 ± 0.02 [d] | 3.77 ± 0.01 [e] |
| | 25 °C/70 rpm | 4.23 ± 027 | 5.29 ± 0.01 [e] | 2.86 ± 0.03 [a] |
| | 25 °C/120 rpm | | 4.37 ± 0.01 | 3.41 ± 0.02 [c] |
| Phenylalanine [3] (Phe) (mg/kg) | 15 °C | | 0.75 ± 0.00 | 1.76 ± 0.01 [c] |
| | 15 °C/70 rpm | 1.03 ± 0.03 | 0.86 ± 0.01 | 0.95 ± 0.02 [a] |
| | 15 °C/120 rpm | | 0.61 ± 0.02 | 1.28 ± 0.04 [b] |
| | 25 °C | | 2.77 ± 0.02 [a] | 2.56 ± 0.03 [e] |
| | 25 °C/70 rpm | 2.56 ± 0.03 | 3.26 ± 0.02 [b] | 1.40 ± 0.04 [b] |
| | 25 °C/120 rpm | | 2.91 ± 0.05 | 2.00 ± 0.04 [c] |
| Lysine (Lys) (mg/kg) | 15 °C | | 97.13 ± 0.13 [b] | 37.26 ± 0.01 [a] |
| | 15 °C/70 rpm | 347.6 ± 8.99 | 191.83 ± 0.56 [f] | 53.38 ± 0.06 [d] |
| | 15 °C/120 rpm | | 121.06 ± 1.67 [c] | 40.92 ± 0.17 [b] |
| | 25 °C | | 72.91 ± 0.08 | 45.67 ± 0.09 |
| | 25 °C/70 rpm | 63.29 ± 0.24 | 67.49 ± 0.15 | 62.04 ± 0.20 |
| | 25 °C/120 rpm | | 66.53 ± 0.09 | 50.94 ± 0.09 |
| Histidine [3] (His) (mg/kg) | 15 °C | | 1.10 ± 0.01 | 4.78 ± 0.01 [c] |
| | 15 °C/70 rpm | 2.20 ± 0.04 | 1.82 ± 0.01 | 4.79 ± 0.00 |
| | 15 °C/120 rpm | | 1.13 ± 0.01 | 3.53 ± 0.00 [a] |
| | 25 °C | | 4.96 ± 0.01 [d] | 5.05 ± 0.03 [d] |
| | 25 °C/70 rpm | 4.96 ± 0.02 | 5.29 ± 0.02 [e] | 5.03 ± 0.03 |
| | 25 °C/120 rpm | | 4.08 ± 0.02 | 5.55 ± 0.00 [e] |
| Arginine (Arg) (mg/kg) | 15 °C | | 157.58 ± 2.44 [b] | 9.05 ± 0.01 [c] |
| | 15 °C/70 rpm | 336.66 ± 10.19 | 285.29 ± 1.90 [c] | 7.24 ± 0.06 [a] |
| | 15 °C/120 rpm | | 162.29 ± 2.91 [b] | 7.95 ± 0.05 [a] |
| | 25 °C | | 12.41 ± 0.01 | 10.78 ± 0.05 [d] |
| | 25 °C/70 rpm | 12.51 ± 0.03 | 11.23 ± 0.01 | 11.89 ± 0.07 [e] |
| | 25 °C/120 rpm | | 10.76 ± 0.01 | 8.95 ± 0.02 [b] |
| Total | 15 °C | | 445.66 | 250.45 |
| | 15 °C/70 rpm | 1040.98 | 776.19 | 294.83 |
| | 15 °C/120 rpm | | 467.23 | 221.04 |
| | 25 °C | | 359.22 | 274.68 |
| | 25 °C/70 rpm | 378.57 | 394.53 | 385.28 |
| | 25 °C/120 rpm | | 274.9 | 361.87 |

Note: [1] indicates umami amino acids (UAA), [2] indicates sweet amino acids (SAA), and [3] indicates bitter amino acids (BAA). Values were obtained from average of all samples and given as means ± SD (n = 5). Different letters indicate significant differences among the treatments.

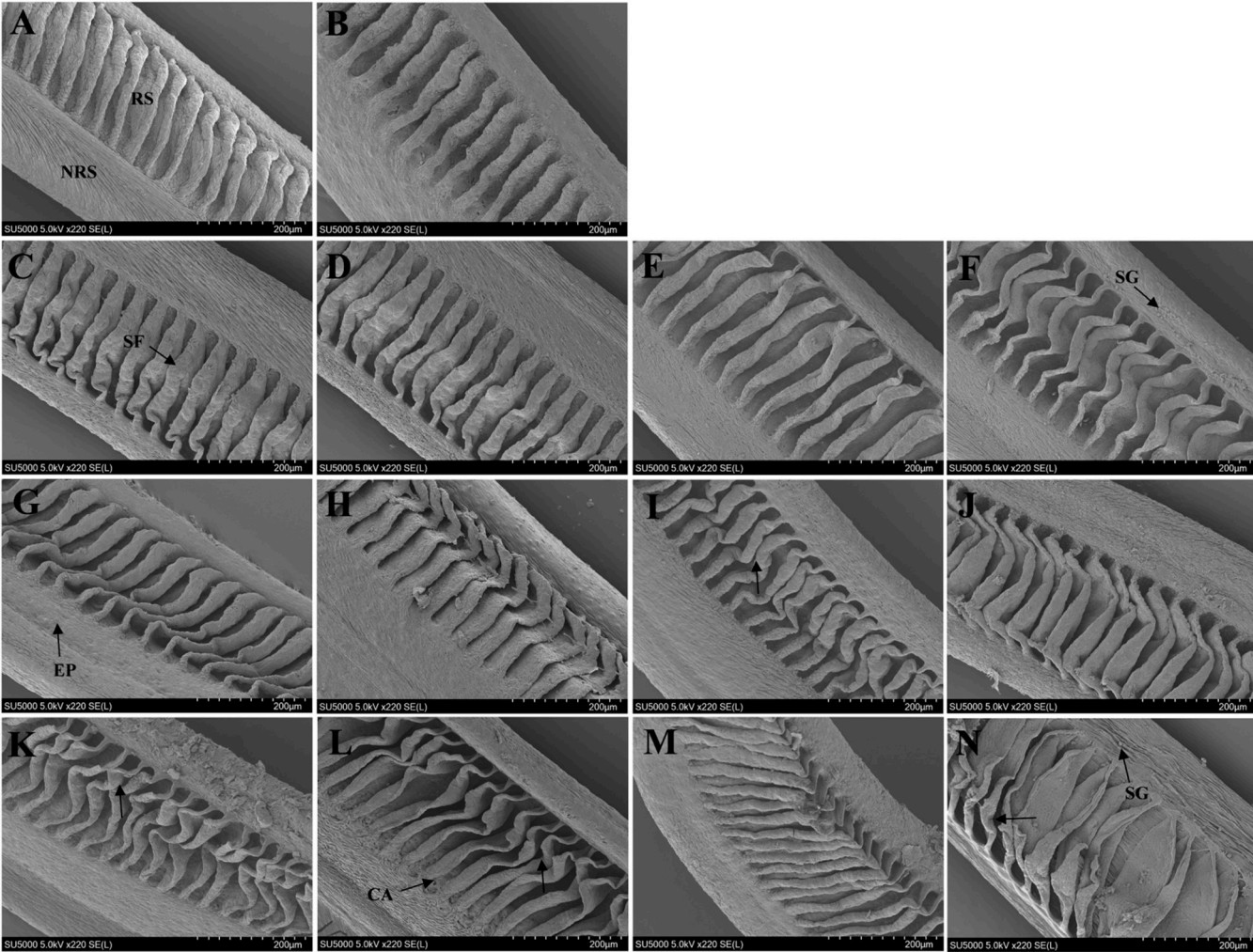

**Figure 7.** Representative SEM micrographs showing changes in gill structure after different temperatures and vibration rates. (**A**) 15 °C, (**B**) 25 °C, (**C**) 24 h—15 °C, (**D**) 24 h—25 °C, I 24 h—15 °C/70 rpm, (**F**) 24 h—25 °C/70 rpm, (**G**) 24 h—15 °C/120 rpm, (**H**) 24 h—25 °C/120 rpm, (**I**) 48 h—15 °C, (**J**) 48 h—25 °C, (**K**) 48 h—15 °C/70 rpm, (**L**) 48 h—25 °C/—70 rpm, (**M**) 48 h—15 °C/120 rpm, and (**N**) 48 h—25 °C/120 rpm. Bar = 200 μm. Non-respiratory surface (NRS), respiratory surface (RS), gill lamellae (GL), secretory granules (SG), epithelial cells (EP), cavities (CA), and disorganized gill lamellae (↑).

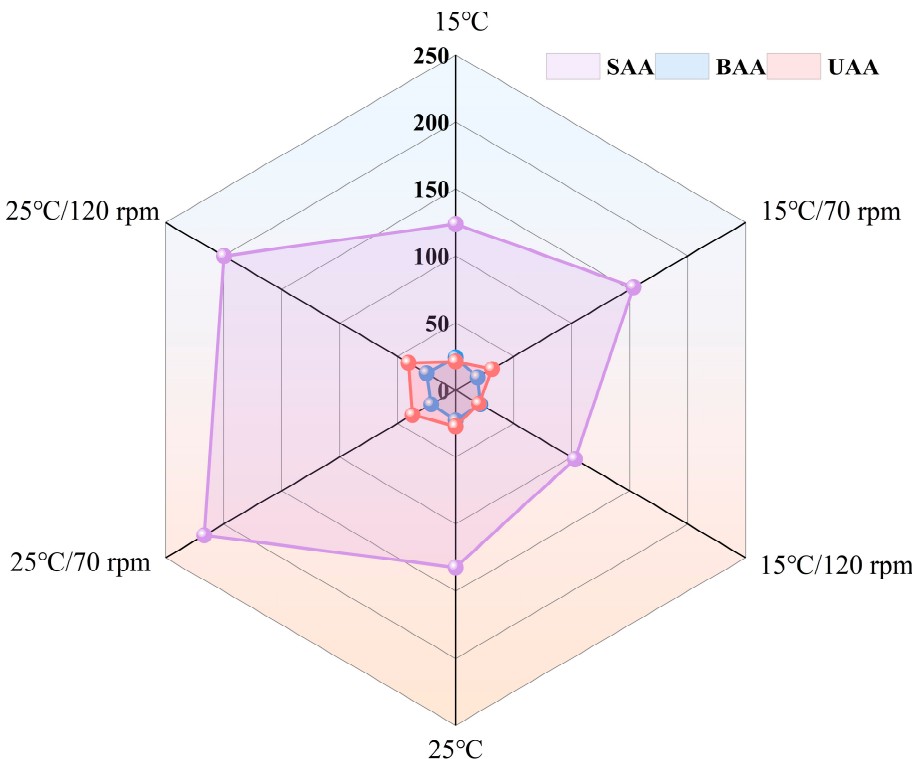

**Figure 8.** Changes in sweet amino acids (SAA), bitter amino acids (BAA), and umami amino acids (UAA) in grouper muscle during 24 h transport.

## 4. Discussion

Water is the main medium for fish to exchange gas and ions, and it is also a purifier for fish metabolites. In this study, water quality parameters changed with increasing transport time and vibration frequency, such as decreased pH, decreased DO, and increased TAN concentration. During closed transport, the increased transport activity led to an increase in the fish's respiration rate [25]. Since carbon dioxide is a by-product of fish respiration, the carbon dioxide produced by respiration dissolves in the water, leading to a decrease in pH [26]. At the same time, the metabolism of fish releases a large amount of unionized ammonia into the water, thereby increasing the concentration of ammonia nitrogen [20]. In the present study, TAN levels increased significantly to 2.44 mg/L within 48 h of transport in the 25 °C/120 rpm group, with a 33% post-transport mortality rate. An increase in ammonia concentration leads to the conversion of $NH_3$ to $NH_4^+$, resulting in a slight increase in the alkalinity of the water and a decrease in the oxygen-carrying capacity [27]. After transporting for 24 h, the high-temperature treatment group showed higher OD values than the low-temperature transport group. This may be because low water temperature effectively reduced the activity and metabolism of fish [28]. Temperature affects the metabolic rate of fish and has a direct impact on fish excretion. The increase in excreted metabolites can cause water pollution and affect the survival rate of fish. In short, deteriorating water quality can affect fish physiology.

AST and ALT are aminotransferases involved in the metabolism of proteins in the body. Serum AST and ALT activities are low and relatively constant under normal conditions. However, when tissue damage and organ dysfunction occur in the body, a large amount of AST and ALT is released into the plasma [29,30]. This study showed that the AST and ALT activities of the transport group showed a trend of first increasing and then decreasing. AST and ALT levels were higher in the transport treatment group than in the control group, and the changes in AST and ALT levels were greater in the high-temperature treatment group within 24 h. It can be seen that the grouper produced a stress response under transport stress, and the liver was damaged. Thereby, increasing the permeability of the

cell membrane, a large amount of AST and ALT is released into the serum, leading to an enhancement in serum AST and ALT activity [2]. Earlier research has demonstrated that transport stress can result in increased levels of AST and ALT in the blood [31]. Therefore, AST and ALT are often used to assess the degree of liver damage [32].

COR is an important glucocorticoid hormone that participates in the regulation of homeostasis and metabolism in the body. Previous studies have shown that an increase in COR content is the main stress manifestation in fish to a stressor [33,34], while an increase in GLU concentration indicates the persistence of the stress response [35]. In this study, the serum levels of COR and GLU in the experimental group were significantly higher than those in the control group during the 12 h of transportation. This indicated that at the initial stage of transportation, the fish body could not adapt to the stimulation and mechanical damage brought by the transportation vibration in time, and different degrees of stress responses resulted in the increase in COR content. This is consistent with previous research results. Xie et al. [36] found that the stress of handling and transport significantly increased plasma COR and GLU concentrations in fish compared to controls. Mirghaed et al. [37] found that serum COR and GLU levels increased after carp transport and then returned to normal levels within 5 h. Transport stress leads to increased levels of COR, which stimulates gluconeogenesis and glycogenolysis. This, in turn, increases the GLU content [38]. In addition, serum COR levels were higher in the high-temperature and high-frequency transport groups than in the low-temperature transport group. This may be because the metabolism of fish in the low-temperature and low-frequency group is weaker and the water quality deteriorates more slowly. The high temperature and high frequency caused the grouper to have a severe stress response. This shows that high temperature and high-frequency vibration are not suitable as a technological condition for grouper water transportation [39]. LDH is the terminal enzyme in sugar metabolism that catalyzes the production of lactate from pyruvate. Myocardial infarction, liver disease, and blood disease can all lead to an increase in LDH [40]. In this experiment, the LDH content of both temperature transport groups showed an overall increase and then decreased to a stable level. This indicated that after the initial transport stress, the anaerobic respiration of carbohydrates in the fish was activated, and a large amount of LDH was released into the blood, resulting in an increase in serum LDH content. After 48 h of transport, the activity returned to the pre-stress levels, indicating that the transport stress damage to the tissues and organs of groupers was short-lived, and the glycolysis process could return to the pre-stress level.

The antioxidant system of the fish organism is in a dynamic equilibrium under normal conditions. When fish are stressed, the organism produces large amounts of reactive oxygen radicals (ROS), and the production of ROS may alter the activity of antioxidant enzymes [41]. Among them, SOD, CAT, and GSH-PX are essential antioxidant enzymes in biological systems. SOD converts harmful superoxide anions into hydrogen peroxide and oxygen, followed by enzymes such as CAT working in concert to convert hydrogen peroxide into oxygen and harmless water [42]. GSH-PX has a protective effect on biofilms. In this experiment, liver SOD, CAT, and GSH-PX were significantly increased compared with the control group, and the enzyme activities almost increased first and then decreased. The elevated SOD activity indicated that the groupers experienced a severe stress response during transportation, resulting in the generation of a large number of free radicals in the body [43]. In order to maintain the stability of the body, the activity of SOD increases rapidly to eliminate excess ROS. The decrease in enzyme activity may be due to the irreversible damage to the body's antioxidant system caused by excessive free radicals. The increase in antioxidant enzyme activity is thought to be a physiological response to the elimination of ROS (Figure 9). GSH-PX and CAT are involved in the removal of hydrogen peroxide [44]. In the pre-transport stage, the enzyme activity of the high-speed treatment group was lower than that of the high-speed treatment group. It can be seen that strong vibrations lead to stronger oxidative stress in the fish. At low speeds, groupers can maintain the balance of the body's antioxidant system by repairing weak damage by itself, which is

more suitable for the water transportation process of grouper. MDA is a product of lipid peroxidation, and its level can reflect the degree of oxidative damage to the body [23]. In this study, MDA levels increased sharply with increasing simulated transmission time and vibration frequency. This shows that high-speed vibration causes severe oxidative stress in fish, generates a large number of free radicals, induces lipid peroxidation, accelerates metabolism in the body, and leads to excessive accumulation of metabolites [45].

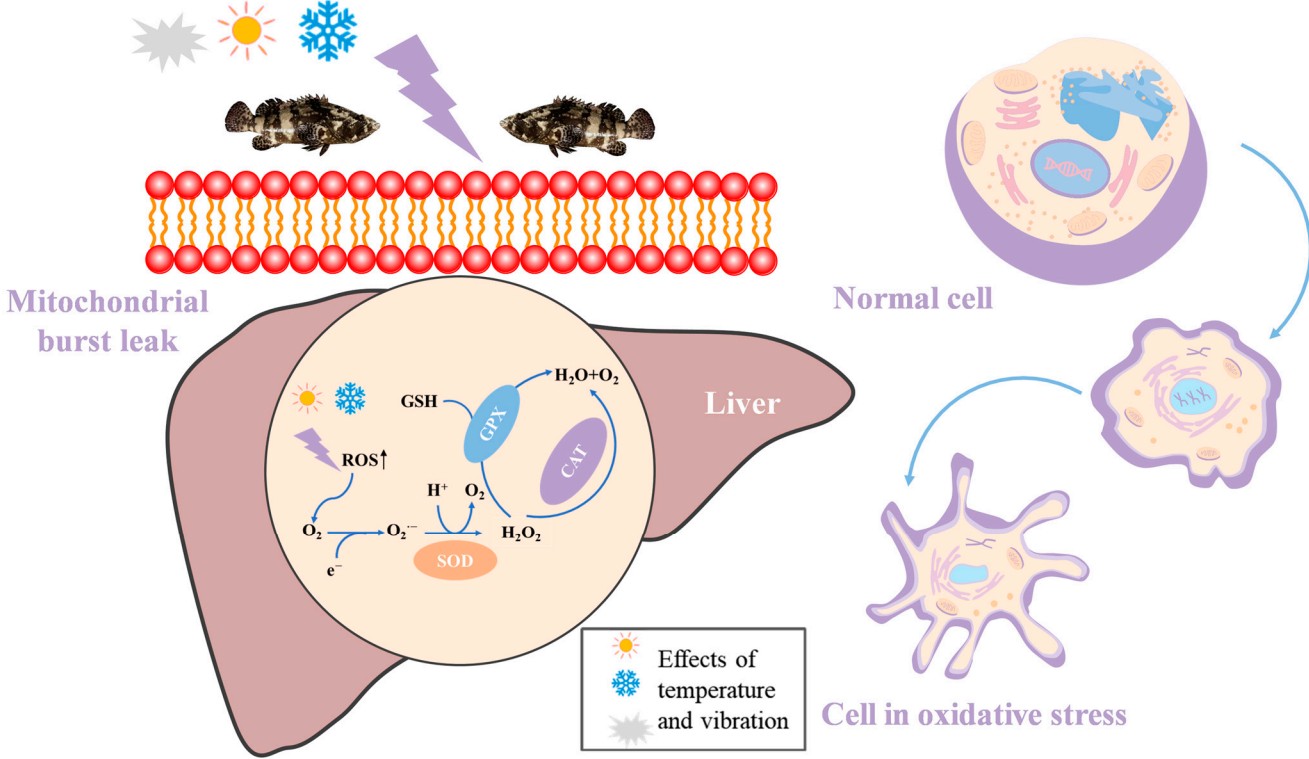

**Figure 9.** The groupers were subjected to temperature and vibration stress, and large amounts of reactive oxygen species were produced in the organisms. The antioxidant enzymes SOD, CAT, and GSH-PX were increased to eliminate the production of reactive oxygen species. Normal cells began to experience apoptosis.

The gills are the main respiratory organs of fish, responsible for excreting metabolic waste and regulating osmotic pressure. The gills are in direct contact with the water, so water stress can have a significant impact on them. Simon Kumar Das et al. [46] found that higher water temperatures increased oxygen consumption and caused gill tissue damage in groupers. In this study, the gill lamellae of the transported fish began to suffer from different degrees of damage, including bending and folding. Later in the transport, the gill lamellae secreted more mucus cells, and the distance between the gill lamellae became shorter. This may be due to the increased breathing area of the gills in fish used to obtain more oxygen [47]. These findings are consistent with changes in gill tissue in sea bass under transport stress [48]. With the increase in transportation time, vibration speed, and temperature, a large number of secretory granules and cavities began to be produced on the gill lamella; the epithelial cells of the gill lamella swelled and fell off; and the gill filaments adhered and fell off each other. This phenomenon may be due to oxidative stress in fish due to environmental stress during transportation. High temperatures and high-frequency vibration accelerate fish metabolism and cause more serious mechanical damage. In addition, ammonia concentrations above a certain threshold in the later stages of fish transport can trigger gill congestion and capillary dilation [49].

FAAs are important components of non-protein nitrogen and play an important role in evaluating the freshness and flavor of fish products. The flavor presentation of FAAs

is related to the hydrophobicity of the side chain R groups. The more hydrophobic Leu, Tyr, His, Ile, and Phe are dominated by bitterness. Glu and Asp with acidic side chains are predominantly fresh-tasting. Ser, Gly, and Thr have less-hydrophobic side chains and are predominantly sweet [50]. In this study, the TFAA content of grouper muscles presented a decreasing tendency after 48 h of transport. The respiratory metabolism and stress response of the samples in the high-temperature and high-speed vibration transportation group were stronger than those in the other groups. This indicated that transport stress depleted the protein, which in turn led to a decrease in TFAA content [51]. In addition, groupers lacked food during transport, which accelerated protein depletion. The FAA results in this study were consistent with the results of water quality deterioration, indicating that the amino acid content was temperature- and speed-dependent following transport stress. There was also a trend towards an increase in SAA and UAA, suggesting that simulated transport could better maintain the freshness of the fish.

## 5. Conclusions

We investigated the blood biochemical parameters, gill histology, oxidative stress, and meat quality changes of pearl gentian grouper under different temperature and rotational speed stresses. Transport stress affected the antioxidant capacity of the fish, damaged gill histological morphology, and accelerated protein depletion. The results of this study also showed that high temperatures and high rotational speeds accelerated the deterioration of water quality, resulting in increased mucus on the surface of fish, swelling of gill epithelial cells, and a stronger oxidative stress response in fish. In contrast, the low-temperature and low-vibration group (15 °C/70 rpm) could better maintain the normal physiological metabolism of organisms under transport pressure. This condition was more conducive to the transport of groupers in water to keep them alive.

**Author Contributions:** Conceptualization, D.F., W.Q. and J.M.; methodology, D.F.; software, D.F.; validation, J.X.; formal analysis, D.F.; investigation, W.Q. and J.M.; resources, J.M.; data curation, D.F.; writing—original draft preparation, D.F.; writing—review and editing, W.Q., J.M. and J.X.; visualization, J.X.; supervision, W.Q., J.M. and J.X.; project administration, J.X.; funding acquisition, J.X. All authors have read and agreed to the published version of the manuscript.

**Funding:** This research was supported by the earmarked fund for CARS-47, and the Shanghai Professional Technology Service Platform on Cold Chain Equipment Performance and Energy Saving Evaluation (20DZ2292200, 19DZ1207503).

**Institutional Review Board Statement:** This study was conducted in accordance with the "Guidelines for Experimental Animals" of the Ministry of Science and Technology (Beijing, China) and approved by the Institutional Animal Care and Use Committee of Shanghai Ocean University (SHOU-DW-2022-103).

**Data Availability Statement:** All data, models, and codes generated or used during the study appear in the submitted article.

**Conflicts of Interest:** The authors declare no conflict of interest.

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
