# Peer review of "The Effects of Transport Stress (Temperature and Vibration) on Blood Biochemical Parameters, Oxidative Stress, and Gill Histomorphology of Pearl Gentian Groupers"

_fishes, doi:10.3390/fishes8040218_

Round 1
Reviewer 1 Report
The authors investigated the effect of transport stress (temperature and vibration) on blood biochemical parameters, oxidative stress, gill histomorphology of pearl gentian grouper.
This manuscript (MS) was clearly written and easy to understand. This work can help the sustainability of this species farming as few studies have been done on this topic. However, some major issues significantly compromised the quality of this MS.
- First, the manuscript needs to be edited by a native English speaker to improve the language of the MS and fix errors.
· Title change to: The effects of transport stress (temperature and vibration) on blood biochemical parameters, oxidative stress and gill histomorphology of pearl gentian grouper
· Line 27, It Is not clear what is 15 °C/120 rpm. Please mention earlier in the abstract that you tested the different temperatures and also shaking.
· Please mention fish size as well in the abstract.
·
· Line 50-52, revise
· Line 79-80 revise.
· Line 97, you said larvae in abstract and fish is 450 gr! Please revise it.
· Line 107 is not clear.
· Transport for how long?
· Figure 1. It only confuses the reader; please delete it, as you did not present it well.
· Figure 2, throughout the MS select a unique colour for 25-degree treatments. The purpose is to be easier for the reader to recognise the treatments.
· The statistical analysis should be updated. You have the effect of temperature, vibration, and time. Three-way Anova can be a great option. When the interaction was significant, please unpack the data in the original one that you already did. If the interaction was not significant, you have to analyse pooled data. You can check this paper and its supplementary file to see how to provide a Table and analyse data. https://onlinelibrary.wiley.com/doi/full/10.1111/anu.13362 Please update the figures and provide a Table summary of the statistical analysis like the paper.
· Figure 3, as was mentioned earlier, please be consistent with colour of the plot for each treatment throughout the MS.
· Figure, when there is no significant difference no need to have a subset (a,b c,,,,) in figures or tables. Without a subset means, there is no significant difference.
· Table 2, please check the data; some of them look repetitive.
· Figure 8, please make sure you mention what these are bittter …
· Please update the results and discussion with the new statistical analysis,
· Here and elsewhere, report P uppercase and italic (P<0.05).
· Throughout the MS, if there is no significant difference, no need to report P-value.
· Please reorder the keywords alphabetically and capitalise each word.
· Please write the abstract more numerically about the results. You can do it by adding their numbers in parentheses.
· Here and throughout the MS, please first mention the common name plus the scientific name, and for the rest of the MS, just report the common name.
· Please update the introduction with recent works as many studies are available from the last two years, which were not included in this section.
· Please mention the novelty of your work in the last paragraph of the introduction.
· For each analysis, please clarify how many fish were taken.
· As a general comment: please focus on fish as hips of the references and studies are available, and no need to cite other vertebrates.
· Although you wrote this section well, you can still improve it by answering these questions and annotating them into the discussion section. Why were these results observed? Discuss more possible reasons.
·
Tables and Figures
• Please explain a little bit about your experimental treatments per each Table and Figure. Each Table and figure should represent enough information separately from the text.
• Double-check the units and titles of all Tables.
When revising your manuscript, please consider all issues mentioned in the reviewers' comments carefully with clear outlines for every change made in response to their comments including suitable rebuttals for any comments you deem inappropriate. Please itemise your response to each review comment, and highlight the revised at re-submission.
Best regards
Author Response
Dear editors,
Thank you for the reviewers’ useful comments, which give us a big help for the later research. The manuscript has been revised accordingly, and the detailed corrections are listed below point by point:
Reviewers' comments:
Q: Title change to: The effects of transport stress (temperature and vibration) on blood biochemical parameters, oxidative stress and gill histomorphology of pearl gentian grouper.
Answer: Title has been revised to read "The effects of transport stress (temperature and vibration) on blood biochemical parameters, oxidative stress and gill histomorphology of pearl gentian grouper". (lines 2-4)
Q: Line 27, It Is not clear what is 15 °C/120 rpm. Please mention earlier in the abstract that you tested the different temperatures and also shaking.
Answer: The meaning of 15 °C/120 rpm has been specified in the abstract. (lines 20-23)
Q: Please mention fish size as well in the abstract.
Answer: The size of the grouper is 450 ± 25 g. (line 19)
Q: Line 50-52, revise
Answer: This sentence has been modified. (lines 47-50)
Q: Line 79-80 revise.
Answer: This sentence has been modified. (lines 72-73)
Q: Line 97, you said larvae in abstract and fish is 450 gr! Please revise it.
Answer: The size of the grouper is 450 ± 25 g and has been modified. (line 88)
Q: Line 107 is not clear.
Answer: The sentence has been added as "The fish were randomly dispersed in 24 double-layered nylon plastic bags (50 cm × 90 cm, 6 fish in each bag, fish to water ratio was 1 : 3). (lines 99-100)
Q: Transport for how long?
Answer: This experiment transports for 48 h. (lines 106-107)
Q: Figure 1. It only confuses the reader; please delete it, as you did not present it well.
Answer: Figure 1 is a schematic diagram of simulated transportation of grouper. The grouper was temporarily raised for 24 h through a chiller and a circulating water tank. The fish were placed in a plastic bag for oxygenation and placed on a vibration table to simulate transportation for 48 h. The figure has been supplemented. (lines 110-113)
Q: Figure 2, throughout the MS select a unique colour for 25-degree treatments. The purpose is to be easier for the reader to recognise the treatments.
Answer: The colors of the pictures for the different temperature groups have been modified. (line 195)
Q: The statistical analysis should be updated. You have the effect of temperature, vibration, and time. Three-way Anova can be a great option. When the interaction was significant, please unpack the data in the original one that you already did. If the interaction was not significant, you have to analyse pooled data. You can check this paper and its supplementary file to see how to provide a Table and analyse data. https://onlinelibrary.wiley.com/doi/full/10.1111/anu.13362 Please update the figures and provide a Table summary of the statistical analysis like the paper.
Answer: The data analysis has been updated based on the literature you provided. Table 1 lists the interactive effects of vibration frequency, temperature, and transit time on free amino acids, oxidative stress, and blood biochemical parameters. However, there are still deficiencies in the statistical analysis of data, and we will learn more in the future. (lines 166-175)
Q: Figure 3, as was mentioned earlier, please be consistent with colour of the plot for each treatment throughout the MS.
Answer: The full text has been revised.
Q: Figure, when there is no significant difference no need to have a subset (a,b c,,,,) in figures or tables. Without a subset means, there is no significant difference.
Answer: The full text has been revised.
Q: Table 2, please check the data; some of them look repetitive.
Answer: The data in the table has been modified. (line 303)
Q: Figure 8, please make sure you mention what these are bittter …
Answer: What are the bitter amino acids, sweet amino acids and umami amino acids are supplemented in the footnote of Table 3. (lines 304-306)
Q: Please update the results and discussion with the new statistical analysis,
Answer: Results and Discussion have been modified.
Q: Here and elsewhere, report P uppercase and italic (P<0.05).
Answer: The full text has been revised.
Q: Throughout the MS, if there is no significant difference, no need to report P-value.
Answer: The full text has been revised.
Q: Please reorder the keywords alphabetically and capitalise each word.
Answer: Keywords have been rearranged and capitalized. (line 37)
Q: Please write the abstract more numerically about the results. You can do it by adding their numbers in parentheses.
Answer: The results have been supplemented in the abstract. (lines 24-35)
Q: Here and throughout the MS, please first mention the common name plus the scientific name, and for the rest of the MS, just report the common name.
Answer: The full text has been revised.
Q: Please update the introduction with recent works as many studies are available from the last two years, which were not included in this section.
Answer: Introduction has been revised.
Q: Please mention the novelty of your work in the last paragraph of the introduction.
Answer: The novelty has been supplemented in the Introduction. (lines 76-80)
Q: For each analysis, please clarify how many fish were taken.
Answer: Five fish were taken from each bag for analysis of blood biochemical parameters, oxidative stress, gill histomorphology, and meat quality. It has been supplemented in the paper. (lines 115-117)
Q: As a general comment: please focus on fish as hips of the references and studies are available, and no need to cite other vertebrates.
Answer: The full text has been revised.
Q: Although you wrote this section well, you can still improve it by answering these questions and annotating them into the discussion section. Why were these results observed? Discuss more possible reasons.
Answer: The conclusion has been supplemented
Q: Please explain a little bit about your experimental treatments per each Table and Figure. Each Table and figure should represent enough information separately from the text.
Answer: Figures and tables have been supplemented.
Q: Double-check the units and titles of all Tables.
Answer: All tables have been checked for units and headings.
The revised manuscript has been resubmitted to the journal. We are looking forward to the positive response.
Yours sincerely,
Weiqiang Qiu and Jing Xie
Reviewer 2 Report
The work presented by Fang, et al. on “The effects of transport stress (temperature and vibration) on 2 blood biochemical parameters, oxidative stress, gill histomorphology in Pearl Gentian grouper” will provide general information regarding different physiological aspects of fish including stress subjected to different weather condition. However, I could see several flaws that need to be addressed. Overall, the English language is poor and needs to be improved.
My comments for improvement of the MS are noted below:
Abstract
Line no. 23-26: At the beginning………. How the activities of the biochemical parameters increase.
The overall applicability of the research findings should be depicted.
In the abstract, the further research areas in the line may also be included.
The Abstract should be re-written.
More keywords should be included like temperature, vibration, etc.
Introduction
Line 54-56: The statements should be omitted
The effects of vibration on transportation stress should be elaborated citing references of the previous works.
Line 67-76: The whole paragraph should be discussed in the discussion part.
Line No. 87-94: The content is needless to be included in the section
The ‘Introduction’ is very extensive which should be crisped.
Clearly mention the aim of the study at the end of the ‘Introduction’
Materials and Methods
Simulated Transportation: How many tanks are used ? How the transportation simulation has been created? Needs more detailed elucidation.
Discussion
Discussion need to be precise and able to extend comprehensive knowledge for further work.
Overall comments:
Overall the research does not bear enough novelty to be published in the reputed journal like ‘Fishes’.
Author Response
Dear editors,
Thank you for the reviewers’ useful comments, which give us a big help for the later research. The manuscript has been revised accordingly, and the detailed corrections are listed below point by point:
Reviewers' comments:
Q: Line no. 23-26: At the beginning………. How the activities of the biochemical parameters increase.
Answer: This section has been revised. (lines 24-30)
Q: The overall applicability of the research findings should be depicted.
Answer: Research conclusions in the abstract have been revised. (lines 24-35)
Q: In the abstract, the further research areas in the line may also be included.
Answer: The abstract has been supplemented。(lines 24-35)
Q: The Abstract should be re-written.
Answer: Abstract has been rewritten. (lines 16-36)
Q: More keywords should be included like temperature, vibration, etc.
Answer: Keywords have been added and reordered. (line 37)
Q: Line 54-56: The statements should be omitted
Answer: This sentence has been deleted.
Q: The effects of vibration on transportation stress should be elaborated citing references of the previous works.
Answer: The literature on the effects of vibration on fish has been supplemented.
Q: Line 67-76: The whole paragraph should be discussed in the discussion part.
Answer: This paragraph has been deleted.
Q: Line No. 87-94: The content is needless to be included in the section
Answer: This paragraph has been removed and modified. (lines :78-83)
Q: The ‘Introduction’ is very extensive which should be crisped.
Answer: The ‘Introduction’ has been revised. (lines 42-85)
Q: Clearly mention the aim of the study at the end of the ‘Introduction’
Answer: The purpose of the study has been added at the end of the introduction. (lines: 80-85)
Q: Simulated Transportation: How many tanks are used? How the transportation simulation has been created? Needs more detailed elucidation.
Answer: A total of 24 boxes were used in the experiment. Put a bag of the same specification in each box. (lines 99-100). The specific experimental process has been supplemented in the text. (lines 98-113)
Q: Discussion need to be precise and able to extend comprehensive knowledge for further work.
Answer: Discussion has been supplemented.
The revised manuscript has been resubmitted to the journal. We are looking forward to the positive response.
Yours sincerely,
Weiqiang Qiu and Jing Xie
Round 2
Reviewer 1 Report
The authors improved the quality of the MS and this is ready to be published. However, please read the MS one more time in the clear version and fix language errors.